# Interactive Planning of Competency-Driven University Teaching Staff Allocation

**Eryk Szwarc [1], Jaroslaw Wikarek [2], Arkadiusz Gola [3,*], Grzegorz Bocewicz [1] and Zbigniew Banaszak [1]**

[1] Faculty of Electronics and Computer Science, Koszalin University of Technology, ul. Śniadeckich 2, 75-453 Koszalin, Poland; eryk.szwarc@tu.koszalin.pl (E.S.); grzegorz.bocewicz@tu.koszalin.pl (G.B.); zbigniew.banaszak@tu.koszalin.pl (Z.B.)

[2] Department of Information Systems, Kielce University of Technology, Al. Tysiąclecia Państwa Polskiego 7, 25-314 Kielce Poland; j.wikarek@tu.kielce.pl

[3] Faculty of Mechanical Engineering, Lublin University of Technology, ul. Nadbystrzycka 36, 20-618 Lublin, Poland

\* Correspondence: a.gola@pollub.pl

**Featured Application: The competency framework for matching student needs and lecturer competencies subject to disruptions caused by teacher absenteeism and curriculum changes. The model is adjusted to perform at the expected robustness level of resultant lecturer allocation. The main strength of the model, as well as the main contribution of this work, consists in that it can absorb the disruptions and produce robust teacher assignment schedules.**

**Abstract:** This paper focuses on a teacher allocation problem that is specifically concerned with assigning available academic lecturers to remaining courses from a given student curriculum. The teachers are linked to tasks according to competencies, competence requirements enforced by the curriculum as well as the number and type of disruptions that hamper the fulfilment of courses. The problem under consideration boils down to searching links between competencies possessed by teachers and competencies required by the curricula that will, firstly, balance student needs and teacher workload and, secondly, ensure an assumed robustness level of the teaching schedule. The implemented interactive method performs iterative solving of analysis and synthesis problems concerned with alternative evaluation/robustness of the competency framework. Its performance is evaluated against a set of real historical data and arbitrarily selected sets of disruptions. The computational results indicate that our method yields better solutions compared to the manual allocation by the university.

**Keywords:** interactive planning; competency framework; teacher assignment; robustness

## 1. Introduction

Allocation problems arise in a range of fields, for example, healthcare, transportation, sports or education, which is a domain whose recognition begins to play an increasingly significant role [1–3]. Everyday practice shows that the organisation and planning of teaching that takes place every year in higher education institutions are associated with the problems of assigning teachers to courses and scheduling/timetabling the courses themselves. Within the education domain, the focal point of Assignment Problems (ASP) is determining the match (or the lack thereof) between the teacher competence (specified by competency sets) and the main subject areas (subsets of curriculum courses) that they are assigned to teach; as such, they are categorised as one of two sub-problems: Allocation (AAP) or Timetabling Problem (TTP) [4–6].

The sub-problems in question can be considered as different types of an assignment problem concerned with the assignment of available resources to various uses. Indeed, AAP can be considered as a type of ASP whose objective is to assign teacher competencies to competencies imposed by curriculum courses. In turn, TTP can be seen as a type of ASP aimed to assign allocations obtained from AAP to a limited number of timeslots and rooms subject to prescribed constraints.

To assess a team of teachers' aptitude to meet the expectations of the curriculum, it is necessary to consider a stock of course requirements and the competencies of the teaching staff members. Teacher competencies are represented collectively as a personnel competency framework [7], which defines current qualifications of staff. It should be noted that course assignment planning decisions (which require specific employee competencies) to resources (teachers with given competencies) take place in dynamically changing organisational settings [8] that involve frequent changes in the scope and structure of objectives, tasks and resources. The examples of such changes include employee absenteeism, loss of qualifications, staff fluctuations and the like. Most of them are random and cannot be anticipated in advance. Such events are henceforth referred to as disruptions [9].

The most frequent reaction to disruption, e.g., caused by employee absenteeism, is to modify the current plan (in this case, the teacher allocation) so as to enable further continuation of the plan [10]. Whether or not the necessary changes can be introduced depends on the competencies of available employees, which may prove insufficient, thus preventing changes in the teacher allocation that would otherwise enable resuming the processing of conducted classes.

In this context, providing for different variants of ASP sub-problems, influenced or not by various disruptions, shall follow the taxonomy of teacher assignment problems given in Figure 1: the Allocation Problem Assuming the Absence of Disruptions (AAPDA), the Timetabling Problem Assuming the Absence of Disruptions (TTPDA), the Allocation Problem Assuming the Presence of Disruptions (AAPDO) and the Timetabling Problem Assuming the Presence of Disruptions (TTPDO). The main objective of both AAPDA and AAPDO problems is the determination of the match (or its lack) between teacher competencies and the main assigned subject areas. The devised competency framework encompassing relationships linking competencies possessed by particular teachers with competencies required by student curriculum is a useful solution for assessing the suitability of teachers with respect to emerging curriculum needs. The considered sub-problems seek the balance between student needs and teacher capabilities, however, in the case of AAPDO, it is additionally required that an assumed robustness level of the resulting teacher allocation is achieved.

In turn, the main objective of TTPDA and TTPDO problems is the determination of the match (or the lack thereof) between allocations linking teachers to curriculum courses and possible space-time assignment constraints, decisive to the occurrence of obstructions. Specifically, their main objective is to assign time and resources to the competency-framework-based allocations so as to satisfy the constraints to the greatest possible extent. Both sub-problems serve to establish timetables guaranteeing that these constraints are fulfilled; however, in the case of TTPDO, the timetable must be accounted for to achieve an assumed robustness level.

In AAPDO and TTPDO problems, the level of robustness is a parameter that describes a competency framework, whose value depends on the type of disruption. The parameter may be expressed as a measure of robustness to the absences of individual teachers. Such a measure provides a decision-maker with a number of absences for which there exists a teacher allocation scheme that guarantees the completion of courses relative to all possible cases of absenteeism.

Since all of the sub-problems distinguished above are based on the same paradigm assuming the balancing of the needs and capabilities, in all of them there may cases that occur in which it should be enough either to reduce expectations (in relation to the possibilities) or to increase possibilities (in relation to expectations).

In that context, the issues of teacher assignment can be approached as analysis or synthesis problems. Considering AAPDA in the former perspective, the question is: Does a competency framework determining the capability of a given teaching staff guarantee a permissible allocation of teachers to curriculum courses?

| Assignment | |
|---|---|
| **Allocation** | **Timetabling** |
| **AAPDA:** The sought links match teacher competencies with competencies required by a student curriculum guaranteeing a balance between student needs and teacher capabilities.<br>**analysis problem**: Does a competency framework guarantee a permissible allocation of teachers to curriculum courses?<br>**synthesis problem**: What changes to the competency framework guarantee a permissible allocation of teachers to curriculum courses? | **TTPDA:** The sought timetables match links (relating a given teacher allocation) and possible space-time location constraints guaranteeing the fulfilment of all constraints.<br>**analysis problem**: Does the teacher allocation guarantee a timetable encompassing the assumed allocation of teachers to curriculum courses while following the imposed time-space constraints?<br>**synthesis problem**: What changes to the teacher allocation guarantees a timetable encompassing an assumed allocation of teachers to curriculum courses while following the imposed time-space constraints? |
| **ASP assuming an absence of disruptions**<br><br>**analysis problem**: Does the competency framework guarantee a timetable encompassing an assumed allocation of teachers to curriculum courses while following the imposed time-space constraints?<br><br>**synthesis problem**: What changes to the competency framework guarantee a timetable encompassing an assumed allocation of teachers to curriculum courses while following the imposed time-space constraints? | |
| **AAPDO:** The sought links match teacher competencies with competencies required by a student curriculum guaranteeing both: a balance between student needs and teacher capabilities, and assumed robustness level of the resulting teacher allocation.<br>**analysis problem**: Does a competency framework guarantee a permissible allocation of teachers to curriculum courses in confronted with given disruptions?<br>**synthesis problem**: What changes to the competency framework guarantee a permissible allocation of teachers to curriculum courses in the case of the occurrence of given disruptions? | **TTPDO:** The sought timetables match links (relating a given teacher allocation) and possible space-time location constraints guaranteeing meeting all the constraints as well as any additional ones imposed by an assumed robustness level of the resulting timetable<br>**analysis problem**: Does the teacher allocation guarantee a timetable encompassing an assumed allocation of teachers to curriculum courses while following the imposed time-space constraints and the occurrence of given disruptions?<br>**synthesis problem**: What changes to the teacher allocation guarantee a timetable encompassing an assumed allocation of teachers to curriculum courses while following the imposed time-space constraints and the occurrence of given disruptions? |
| **ASP assuming occurrence of disruptions**<br><br>**analysis problem**: Does the competency framework guarantee a timetable encompassing an assumed allocation of teachers to curriculum courses while following the imposed time-space constraints and the occurrence of given disruptions?<br><br>**synthesis problem**: What changes to the competency framework guarantee a timetable encompassing an assumed allocation of teachers to curriculum courses while following the imposed time-space constraints and the occurrence of given disruptions? | |

*Left vertical labels: Disruptions; Undisrupted; Disrupted e.g. absenteeism, staff fluctuations, curriculum changes.*

**Figure 1.** Taxonomy of teacher assignment problems.

Whereas, in the synthesis approach, the question is: What changes to the competency framework guarantee a permissible allocation of teachers to curriculum courses? The next case of the above-mentioned problems, TTPDA, seeks to answer similar questions: Does the teacher allocation guarantee a timetable encompassing an assumed allocation of teachers to curriculum courses while

following the imposed time-space constraints? What changes to the teacher allocation guarantee a timetable encompassing an assumed allocation of teachers to curriculum courses while following the imposed time-space constraints? The last question is directly relevant to the synthesis of a competency framework robust to disruptions.

A similar observation highlighting synthesis and analysis problems applies to AAPDA and TTPDA as well as to AAPDO and TTPDO problems. Since ASP, assuming the absence of disruptions, can be regarded as a problem focused on the determination of the match (or its lack) between the results of AAPDA and TTPDA problems, its objective concerns solutions to both the analysis and synthesis aspects of ASP. In the first case, the question asked is: Does a given competency framework providing the allocation of teachers to curriculum courses guarantee a timetable following the imposed time-space constraints? In the second case, regarding the synthesis problems, the question that requires consideration is: What changes to the competency framework guarantee a timetable encompassing an assumed allocation of teachers to curriculum courses while following the imposed time-space constraints? Certainly, the ASP assuming the occurrence of disruptions can be considered in a similar way.

In the context of the taxonomy of assignment problems presented above, further considerations are limited to the class of AAPDO problems, in particular, accentuating versions of the synthesis. Careful attention is devoted to the assessment of the possibility of choosing constraint-programming-based software packages enabling decision-making in the cases of large-scale allocation problems.

The study reported in this paper is a continuation of our previous work that explored methods of rapid prototyping of workforce-allocation solutions and personnel scheduling problems robust to the disruptions occurring in the course of the execution of multiple projects [11–13]. The main contributions of this paper are given below:

1.  In contrast to the common trend of dealing with personnel allocation and staff scheduling problems separately, we propose an integrated approach to the prototyping of robust competency-driven teacher allocations. This approach is based on the taxonomy of human resource allocation problems, which sets the context for the construction of realistic proactive strategies for the robust allocation of teachers.
2.  The solution is a declarative modelling-driven approach to the assessment of alternative variants of a competency framework robust to a selected set of anticipated types of personnel absence. The model searches for competency-driven teacher allocations robust to teacher absences under the Constraint Satisfaction Problem.
3.  Our approach shows the necessary qualities to replace typical operational research or computer simulation methods for workforce allocation and personnel scheduling with constraint programming-driven techniques. Its main advantage is that it accounts for the alternate usage of synthesis and analysis versions of APS, assuming the occurrence of disruptions in the course of planning solutions robust to teacher absenteeism.

The remainder of this paper is organised as follows. An overview of the relevant literature is provided in Section 2. Section 3 describes the example that introduces readers to the competency-driven teacher allocation approach aimed at prototyping competency frameworks under given robustness constraints. Moreover, this section focuses on the generation of robust plans and schedules as well as the measurement of their robustness. In Section 4, of the presented case of course casting provides a representation of university staff allocation planning employing our approach. In addition, this section reports on the computational experiments performed in the LINGO environment (Lindo Systems, 1415 North Dayton Street Chicago, IL, 60642, USA) that verify and illustrate the potential application of the approach to generate competency frameworks robust to the anticipated employee absence. The conclusions and directions for further research are discussed in Section 5.

## 2. Literature Review

In accordance to the literature, AAPDO can be classified into a group of Human Resource Allocation Problems (HRAP), which is part of the generic Resource Allocation Problem (RAP), defined as the process of allocating resources between various projects [14–21]. A resource—a person, material or capital—undergoes the RAP process, which seeks an optimal allocation of a limited amount of resources to a number of tasks by optimising their objectives subject to given resource constraints [22]. Our study focuses on Human Resources (HR) and related HRAP, which is often the source of various problems: production, health care, project management, maintenance staff, hospitality and tourism industries, sport management, reviewer assignment, military and the focal point of this analysis—education [22].

The literature on HRAP in the education domain [23–26] shows that the teaching process in higher education institutions is associated with two problems: the Teacher Allocation Problem (TAP) and the Timetabling Problem (TTP). Although typically it is TAP that precedes TTP [24,25,27–29], several studies [30] indicate that a reverse approach can also be applied. In this case, the main difficulty lies in matching teacher preferences regarding courses or contact hours and their teaching competencies to the existing schedule. Other approaches employ TAP and TTP simultaneously [4,31]. Another solution [32] is a three-step HRAP approach: teachers are assigned to courses (TAP), next, courses are assigned to time slots, and finally, time slots are assigned to classrooms. To date, various problems have been studied with respect to the type of constraints and the type of educational establishment (pre/mid/high-school, university) [23,33–36]. In this context, AAPDO is related to multi-factor TAP, including, competence [37,38], efficiency [39], preferences and availability [40], learning characteristics [41], etc. Numerous studies show (e.g., Reference [42]) that the key factor in this matter is the ability of employees to execute tasks according to the declared competence.

There are multiple definitions of "competence" [43–46]. The understanding of the term has not been consistent, as it tends to be understood as proficiency, ability, capability and capacity. The confusion in the nomenclature has been the subject of some scientific debate and there are works that attempted to set it in order [47]. Their main conclusion is that competence is typically understood as a combination of knowledge, skills, experience and abilities that are described in terms of specific behaviours and are demonstrated by superior performers in a specific job or work role in a particular organisation. In other words, competence describes whether a given worker is able to execute a given task. Consequently, employee competence represented collectively is the competency framework, which can be identified with the competency matrix [48], a commonly used tool in the construction, production and other industries (it defines current qualifications of staff members and their allocation to specific tasks).

To date, numerous models of TAP have been proposed. They have been utilised to minimise the lesson preparation time (teachers being allowed different course preparation times) [49]. In another work [35], a problem with the set of triplets (course, teacher, a group of students) and the set of pairs (period, day) was solved. Other considered problems included the allocation of teachers to courses so as to balance the teacher workload [50]. Authors take into account various requirements following from the specific character of a given university or legislative norms in force in a given country. However, certain parameters exhibit higher uniformity and are widely applicable regardless of the national or legislative context, such as, specifically, not every teacher is qualified to do every course (only competent teachers can be assigned to courses), and each teacher may be assigned to more than one course. In this context, and according to the taxonomy in Pentico [51], three groups of HRAP models are distinguished:

- Assigning at most one task per agent (a one-to-one assignment).
- Assigning multiple agents to a task or assigning multiple tasks to the same agent (a one-to-many assignment).
- Multidimensional assignment problems assigning the members of three or more sets, such as matching jobs with workers and machines or assigning students and teachers to classes and time slots.

While TAP models are categorised under the second HRAP group, further research by Bouajaja and Dridi [22] determined several variants of HRAP models (with mathematical formulations) and assigned them to these three groups.

It should be noted that planning decisions regarding the teacher allocation to courses are made in variable environment settings [8], which involve frequent changes in the scope and structure of courses and teachers. Examples of such variations include teacher absenteeism (sick leave, accidents, maternity leave, etc.), changes in the curriculum, fluctuations of teaching staff, and the like. Such unplanned events constitute disruptions [10]. Issues related to predicting the effects of disruptions, as well as planning competency frameworks and employee allocation robust to disruptions are included in HRAP. Solutions in this area are mostly limited to the introduction of time buffers [52–54] or capacity (potential/resource) buffers [55–57]. The studies, however, fail to provide a quantitative assessment of the impact of the staff competencies on the continuity of the processes carried out in an organisation. As a result, there is a distinct paucity of research on models that would allow building staff teams whose competency frameworks could ensure uninterrupted task realisation.

Given the inherent complexity, solving HRAP, AAPDO or TAP problems would be impossible but for the computational techniques. Human Resource Information System (HRIS) is a software solution developed to aid Human Resource Management (HRM) (within the problems in question). HRIS has been applied [58] to aid the decision-making process in the analysis and planning of human resources, which included such spheres as manpower requirement, skill/competence requirement, absenteeism analysis, placements, job matching, job descriptions, workforce utilisation, payroll, safety training, maintenance of accident records, etc. HRIS also enables selecting the right person at the right time and assigning them to the right job. According to the classification of HRIS functionalities presented in the work of Nawaz and Gomes [59], questions (analysis/synthesis) related to AAPDO/AAPDA (see Figure 1) are associated with selected HRIS areas: HR acquirement (recruitment), HR planning (rostering/scheduling/staffing) and HR development (qualification management). Furthermore, in the context of competency-driven planning staff allocation, these areas fall into the domain of Competence Management Systems (CMS). From the literature on CMS, they emerge as multidimensional and comprehensive approaches, including tools such as competence management, skills-gap analysis, succession planning, competence analysis and profiling [60]. While commercially available CMS programmes share certain common features, none of them contain disruption occurrence features (e.g., employee absenteeism). Thus, the questions related to the analysis and the synthesis of competency framework robustness cannot be answered so far.

The implementation of the newly developed method in, e.g., ERP systems, will enable early identification of needs and quick prototyping of alternative decisions in the area of staff competency management [61]. This solution will allow managers to make personnel decisions online in response to employee absenteeism and/or staffing fluctuation, legislative changes, changes in the scope of orders, etc. It will also enable the development of other, derivative HRM methods, such as methods of supporting the organisation and teamwork planning in situations in which the available workers have to step in for the absent colleagues.

## 3. Competency-Driven Staff Allocation

### 3.1. Problem Formulation

In the general case, the teacher allocation problem boils down to allocating resources (teachers) to activities (courses). A well-prepared allocation must guarantee the satisfaction of constraints related to the specific contact hour limits, minimum academic staff complement, courses taught by instructors with an appropriate academic degree or title, etc. In the analysis presented below, the formal definition of the Teacher Allocation Problem (TAP) is adopted. It accentuates the role of the competency framework of the available teaching staff.

Given is a set of courses to teach in a given academic year. The courses are conducted in the winter or the summer semester. There are various course formats (lectures, tutorials, labs, workshops, seminars) and each course (taught in a specific format) has a fixed number of course sections or classes. Students in each course section are obliged to attend a fixed number of course meetings (colloquially called classes). Each course meeting has an assigned number of teaching credits.

Also given is a set of teachers. Each teacher (lecturer) has a set of competencies (skills, qualifications) to teach specific courses/course meetings. The set is described in a binary way (can/cannot teach a specific course). The competencies of individual teachers make up the competency framework of the entire staff (a set of teachers).

The staff members' competencies can change. The resulting change in the framework is understood as an acquisition, by at least one staff member, of new competencies, allowing them to teach a specific course (a group of courses).

In addition, each teacher has an academic/professional degree/title (professor, doctor, master, engineer) and is assigned a specific number of teaching credits (teaching load, credit hours) per academic year.

A permissible teacher allocation is understood as an allocation of teachers to course meetings which satisfies the following constraints:

- Each course meeting can be conducted by only one competent teacher,
- The set of all course meetings for a specific course section (group) can be run by no more than three teachers,
- Lectures and seminars can only be delivered by professors and doctors,
- Each teacher should have a guaranteed teaching quota,
- All course meetings must be assigned to teachers,
- Others (following from various disruptions and/or individual needs of a university/organisational unit).

The presented assumptions constitute the verbal model of the ASP problem, in which detailed sub-problems are formulated in accordance with the adopted taxonomy, see Figure 1. Indeed, focusing on AAPDO, the problem of synthesis can be considered in the context of searching for the competency framework, $G$, robust to disruptions that affect the possibility of carrying out the ordered activities.

This situation is illustrated by the following simplified example. Given is the set of courses containing three courses: $Z = \{Z_1, Z_2, Z_3\}$. Each course, $Z_i$, has a number of tasks assigned to it. Consequently, courses $Z_1$ and $Z_2$ require unit tasks, whereas course $Z_3$ requires 2 tasks. A staff of three teachers from set $\mathcal{P} = \{P_1, P_2, P_3\}$ is planned for the execution of the courses. The competencies of the individual teachers comprise the personnel competency framework presented in Table 1. The framework is represented by a competency matrix, $G$, in which values "1" and "0" stand for having or not having a specific competency, respectively.

The organisation observes the following course completion rules (constraints):

(a) A task of a course can only be carried out by a competent teacher,
(b) Each teacher must perform no less than 1 task and no more than 2 tasks,
(c) All tasks of courses must be assigned to teachers.

**Table 1.** Competency framework, $G$.

| $G$ | | Competencies Enabling the Execution of Courses, $Z_i$ | | |
|---|---|---|---|---|
| | | $Z_1$ | $Z_2$ | $Z_3$ |
| Teachers | $P_1$ | 1 | 1 | 0 |
| | $P_2$ | 0 | 0 | 1 |
| | $P_3$ | 1 | 1 | 0 |

The competency framework enables the completion of tasks in a process that satisfies the constraints above, in accordance with teacher allocation, $X$, presented in Table 2. In a permissible

teacher allocation. $X$, teacher $P_1$ carries out a task of course $Z_1$, teacher $P_3$ carries out a task of course $Z_2$, and teacher $P_2$ carries out two tasks of course $Z_3$.

**Table 2.** A permissible teacher allocation, $X$.

| $X$ | | Number of Tasks of the Course | | |
|---|---|---|---|---|
| | | $Z_1$ | $Z_2$ | $Z_3$ |
| Teachers | $P_1$ | 1 | 0 | 0 |
| | $P_2$ | 0 | 0 | 2 |
| | $P_3$ | 0 | 1 | 0 |

Let us consider a situation in which teacher $P_2$ is absent from the execution of tasks (teacher absence). In this situation, it is necessary to modify teacher allocation, $X$ (Table 2), by substituting the teacher allotted to performing tasks of course $Z_3$. As competency framework $G$ (Table 1) shows, none of the remaining teachers has the competencies to complete tasks of course $Z_3$. The absence of teacher $P_2$ requires that two tasks of course $Z_3$ should be delegated to other teachers. It is not possible to delegate both tasks to them because they lack competencies enabling the execution of course $Z_3$. In this case, the following question should be considered: Does there exist a competency framework, $G$, that guarantees a permissible allocation, $X$, despite the absence of teacher $P_2$? This competency framework, $G$, is presented in Table 3. Permissible teacher allocation, $X$, where one task of course $Z_3$ is allocated to teacher $P_1$ and one to teacher $P_3$ is presented in Table 4. In this situation, constraints (b) and (c) are satisfied (Table 4).

**Table 3.** Competency framework, $G$.

| $G$ | | Competencies Enabling the Execution of the Courses, $Z_i$ | | |
|---|---|---|---|---|
| | | $Z_1$ | $Z_2$ | $Z_3$ |
| Teachers | $P_1$ | 1 | 1 | 1 |
| | $P_2$ | 0 | 0 | 1 |
| | $P_3$ | 1 | 1 | 1 |

**Table 4.** A permissible teacher allocation, $X$, in the case of the absence of teacher $P_2$.

| $X$ | | Number of Tasks of the Course | | |
|---|---|---|---|---|
| | | $Z_1$ | $Z_2$ | $Z_3$ |
| Teachers | $P_1$ | 1 | 0 | 1 |
| | $-$ | - | - | - |
| | $P_3$ | 0 | 1 | 1 |

It is easy to note that the obtained competency framework, $G$, presented in Table 3, ensures that tasks are completed despite the absence of teacher $P_2$, which means the framework is robust to the type of disruption under consideration.

In the example considered above, competency framework, $G$ (Table 3), guarantees permissible teacher allocation, $X$, regardless of which teacher is absent. In other words, the competency framework, $G$, is fully robust to the absence of a single teacher. This situation is not always possible. Typically, the given competency framework will secure only certain variants of teacher absence. The number of scenarios for a given disruption for which a given competency framework guarantees an acceptable allocation of teachers to courses is determined by what is known as the robustness of a competency framework function.

Describing the robustness of competency framework, $G$, of a given staff of teachers, $\mathcal{P}$, who perform tasks of set $\mathcal{Z}$ (a set of courses), to the absence of $\omega$ teachers requires the concept of Robustness of a Competency Framework, that is defined as a function $R_{\mathcal{P}}^{\mathcal{Z}}: \Omega \to [0,1]$, where: $\Omega = \{1,..,|\mathcal{P}|\}$, a domain of variable $\omega$ describing the number of absent teachers. Therefore, the function of the competency framework robustness is determined from the equation:

$$R_{\mathcal{P}}^{\mathcal{Z}}(\omega) = \frac{|LP_{\omega}|}{|U_{\omega}|} \tag{1}$$

where:

$U_{\omega}$    is a family of $\omega$-number of simultaneous teacher absences: $U_{\omega} = \left\{ u_i | u_i \subseteq \mathcal{P}; |u_i| = \omega; i = 1 \dots \binom{|\mathcal{P}|}{\omega} \right\}$, $\omega \in \{1, \dots, |\mathcal{P}|\}$; for example, for the case of an absence of two teachers in the example above (Tables 3 and 4), the set $U_2 = \{\{P_1, P_2\}, \{P_1, P_3\}, \{P_2, P_3\}\}$ contains 3 absence scenarios.

$LP_{\omega}$    is a subset of set $U_{\omega}$ ($LP_{\omega} \subseteq U_{\omega}$) which contains absence scenarios for which competency framework, $G$, guarantees permissible teacher allocation, $X$, to courses in the event of absences of teachers. In a special case, subset $LP_{\omega}$ can be an empty set, i.e., $LP_{\omega} = \emptyset$, which corresponds to a situation in which no suitable replacement enabling the execution of all the courses from set $\mathcal{Z}$ can be found for any of the 28 absence scenarios.

The adopted measure of robustness, $R_{\mathcal{P}}^{\mathcal{Z}}(\omega)$, which is a ratio between the number of absences for which there are appropriate replacements $|LP_{\omega}|$ to the number of all possible cases of absence $|U_{\omega}|$, takes values from the range $[0,1] \subset \mathbb{R}$, where:

- $R_{\mathcal{P}}^{\mathcal{Z}}(\omega) = 0$ stands for lifelessness (robustlessness), i.e., there is no suitable replacement enabling the execution of all tasks from set $\mathcal{Z}$ for any of the possible cases of absence ($\omega$-number of simultaneous teacher absences).
- $R_{\mathcal{P}}^{\mathcal{Z}}(\omega) = 1$ stands for full robustness, i.e., for each of the possible cases of absence ($\omega$-number of simultaneous teacher absences) there is at least one replacement guaranteeing the execution of all the tasks from set $\mathcal{Z}$.

In the context of the proposed measure of robustness, $R_{\mathcal{P}}^{\mathcal{Z}}(\omega)$, the problem of synthesis of the robust competency framework, $G$, boils down to the question: Does there exist, for a given set of tasks, $\mathcal{Z}$, executed by $\mathcal{P}$ teachers, a competency framework, $G$, which guarantees robustness, $R_{\mathcal{P}}^{\mathcal{Z}}(\omega)$ (to the absence of $\omega$ teachers), greater than or equal to arbitrarily adopted values, ${}^{*}R_{\mathcal{P}}^{\mathcal{Z}}$? The search for competency frameworks, $G$, that guarantee a specific level of robustness, $R_{\mathcal{P}}^{Q}(\omega)$, is a combinatorial optimisation NP-hard problem belonging to the class of synthesis problems [12].

In general, the synthesis problem of competency frameworks robust to a selected set of disruptions, AAPDO, can be formulated as follows: Given is a university employing academic staff described by the competency framework provided. The organisation's objectives and the set of tasks it carries out are known. The goal is to find an answer to the question: Does there exist a model and a method of constructing competency frameworks robust to selected disruptions caused by teacher absenteeism, loss of qualifications, etc.?

*3.2. Model*

The declarative framework-driven reference model of AAPDO presented below allows, in a natural way, for the formulation of a constraint satisfaction problem to generate competency frameworks robust to disruptions caused by teacher absenteeism.

A university potential and the student curricula requirements can be represented as a model linking two main elements: set of jobs, $Z$ (courses), and competency framework, $G$, representing the competencies possessed by the university teachers.

Set of courses $\mathcal{Z}$: The courses determined by student curricula are represented by set $\mathcal{Z} = \{Z_1, \dots, Z_i, \dots, Z_n\}$, where $Z_i$ is the $i$-th course. It is generally assumed that lectures on a given subject (course $Z_i$) are conducted for a number of student groups. The implementation of course $Z_i$ is, therefore, related to the implementation of a certain number of tasks ($q_i$) of given durations (sizes) ($l_i$). Tasks of $Z_i$ can have different sizes $\left\{ l_{i_1}, l_{i_2}, \dots, l_{i_{q_i}} \right\}$, however, for simplicity, let us assume that each task of $Z_i$ is of the same size, i.e., $l_{i_1} = l_{i_2} = \cdots = l_{i_{q_i}} = l_i$. In that context, course $Z_i$ is described by two elements: the number of tasks ($q_i$) and their duration ($l_i$):

$$Z_i = (q_i, l_i) \tag{2}$$

where: $q_i$ is the number of tasks of course $Z_i$, and $l_i$ is the duration (size) of each task of course $Z_i$.

It is assumed that courses $Z_i$ are conducted by a competent teacher.

University teaching staff: A staff of teachers, $\mathcal{P}$, employed by the university is allocated to perform the planned tasks. Set $\mathcal{P} = \{P_1, \dots, P_k, \dots, P_m\}$ defines a staff of teachers, where $P_k$ is a pair:

$$P_k = (s_k, z_k) \tag{3}$$

where: $s_k$ is the minimum working hours of the $k$-th teacher $(s_k \in \mathbb{N})$, and $z_k$ is the maximum working hours of the $k$-th teacher $(z_k \in \mathbb{N})$.

University teaching staff, $\mathcal{P}$, corresponds to the competency framework defined as a function $G: \mathcal{P} \times Z \to \{0,1\}$ that assigns to each pair $(P_k, Z_i)$ a value from the set $\{0,1\}$: 0—when teacher $P_k$ does not have the competence to execute tasks from course $Z_i$, and 1—when teacher $P_k$ has the competence to execute tasks from course $Z_i$. To simplify the notation, it is assumed that $G(P_k, Z_i) = g_{k,i}$. As a result, the matrix representation of the competency framework of a staff of teachers, $\mathcal{P}$, takes the form:

$$G = \left[g_{k,i}\right]_{k=1\dots m; i=1\dots n} \tag{4}$$

where: $g_{k,i} \in \{0,1\}$,

$$g_{k,i} = \begin{cases} 1 & \text{when teacher } P_k \text{ has the competencies to execute tasks from course } Z_i \\ 0 & \text{in remaining cases.} \end{cases}$$

Teacher allocation, $X$, specifies which tasks are assigned to each member of staff, $\mathcal{P}$, in the execution of tasks from course $\mathcal{Z}$. The allocation is a function $X: \mathcal{P} \times Z \to \mathbb{N}$ that assigns to each pair $(P_k, Z_i)$ a value from set $\{0,1,\dots,q_i\}$: 0—when teacher $P_k$ does not execute tasks of course $Z_i$, and $1, \dots, q_i$—when teacher $P_k$ executes tasks of course $Z_i$. For simplicity considerations, it is assumed that $X(P_k, Z_i) = x_{k,i}$, and the resulting matrix representation of allocation takes the form:

$$X = \left[x_{k,i}\right]_{k=1\dots m; i=1\dots n} \tag{5}$$

where: $x_{k,i} \in \{0,1,\dots,q_i\}$ represents the number of tasks from course $Z_i$ executed by teacher $P_k$.
It is further assumed that:

- Tasks can only be executed by a competent teacher, i.e., $\forall_{k=1\dots m, i=1\dots n} x_{k,i} \leq q_i \times g_{k,i}$.
- Teacher working time limits may not be exceeded, i.e., $\forall_{k=1\dots m, i=1\dots n} \sum_{i=1}^{n} x_{k,i} \times l_i \geq s_k$ and $\sum_{i=1}^{n} x_{k,i} \times l_i \leq z_k$.

Disruptions and the measure of robustness of the competency framework: Let us discuss one type of disruption, namely teacher absences. This disruption is characterised by a family of $\omega$ combinations of set: $U_\omega = \left\{u_i | u_i \subseteq \mathcal{P}; |u_i| = \omega; i = 1 \dots \binom{|\mathcal{P}|}{\omega}\right\}$. In other words, the family $U_\omega$ consists of scenarios parametrised by $\omega$—the number of simultaneous teacher absences. The occurrence of this type of disruptions spurs the search for allocation, $X$, that allows a set of courses, $Z$, to be executed without interruptions. How this should be interpreted is that when one teacher is absent, their responsibilities can be transferred to another currently available teacher. It is not always possible to implement such an allocation, $X$, however, to assess the chances of implementing an allocation, we use the concept of robustness of a competency framework (1).

To put this type of problems into formal terms, the following reference model is introduced:
Sets:

$\mathcal{Z}$:　　Set of courses: $\mathcal{Z} = \{Z_1, \dots, Z_i, \dots, Z_n\}$

$\mathcal{P}$:　　Set of teachers: $\mathcal{P} = \{P_1, \dots, P_k, \dots, P_m\}$

$U_\omega$:　　Family of scenarios parametrised by $\omega$—the number of simultaneous teacher absences:
$$U_\omega = \left\{ u_i \mid u_i \subseteq \mathcal{P}; |u_i| = \omega; i = 1 \dots \binom{|\mathcal{P}|}{\omega} \right\}.$$

$\Theta$:　　A single scenario of absence $\omega$ of teachers, $\Theta \in U_\omega$

$LP_\omega$:　Subset $U_\omega$ $(LP_\omega \subseteq U_\omega)$, which contains absence scenarios for which competency framework, $G$, guarantees a permissible teacher allocation, $X$, to courses in the event of absences of teachers.

Parameters:

$n$:　　Number of courses ($n \in \mathbb{N}$)

$q_i$:　　Number of tasks from course $Z_i$

$m$:　　Number of teachers ($m \in \mathbb{N}$)

$\omega$:　　Number of absent teachers, $\mathcal{P}$ ($\omega \in \mathbb{N}$), $\omega < m$

$l_i$:　　Duration of the task from course $Z_i$ (in hours)

$s_k$:　　Minimum working hours of teacher $P_k (s_k \in \mathbb{N})$

$z_k$:　　Maximum working hours of teacher $P_k$ ($z_k \in \mathbb{N}$)

$^*R_\mathcal{P}^Z$:　Predicted robustness of the competency framework ($^*R_\mathcal{P}^Z \in [0,1]$)

Decision variables:

$G$:　　Competency framework given by matrix $G = [g_{k,i}]_{k=1\dots m; i=1\dots n}$, where: $g_{k,i} \in \{0,1\}$:

$$g_{k,i} = \begin{cases} 1 & \text{when teacher } P_k \text{ has the competencies to execute task from course } Z_i \\ 0 & \text{in remaining cases.} \end{cases}$$

Robustness of competency framework, $G$, for the absence of $\omega$ teachers is described by function $R_\mathcal{P}^Z(\omega)$ (1).

$G^\Theta$:　Competency framework which takes into account absences of teachers defined in set $\Theta$:
$$G^\Theta = [g_{k,i}^\Theta]_{k=1\dots m; i=1\dots n} \quad \text{where:}$$

$$g_{k,i}^\Theta = \begin{cases} 1 & \text{when } k \notin \Theta \text{ and } P_k \text{ have the competence to execute task of } Z_i \\ 0 & \text{in remaining cases.} \end{cases}$$

$X$:　　Teacher allocation, $X = [x_{k,i}]_{k=1\dots m; i=1\dots n}$, where: $x_{k,i} \in \{0,1,\dots,q_i\}$ means the number of tasks from course $Z_i$ executed by teacher $P_k$.

$X^\Theta$:　Allocation in situations when teachers defined in set $\Theta$ are absent from work: $X^\Theta = [x_{k,i}^\Theta]_{k=1\dots m; i=1\dots n}$, where: $x_{k,i} \in \{0,1,\dots,q_i\}$ represents the number of tasks from course $Z_i$ executed by teacher $P_k$.

$c^\Theta$:　A variable that specifies whether there exists allocation $X^\Theta$ ensuring execution of tasks from courses, $\mathcal{Z}$. The value of variable $c^\Theta \in \{0,1\}$ depends on ancillary sub-variables: $c_{1,i}^\Theta$, $c_{2,k}^\Theta$, $c_{3,k}^\Theta$, which specify whether constraints (6)–(13) are satisfied.

Constraints:

1.　The element $g_{k,i}^\Theta$ of matrix $G^\Theta$ that characterises the absence of teacher $P_k$ ($P_k \in \Theta$) takes the value 0:

$$g_{k,i}^\Theta = \begin{cases} g_{k,i} & \text{when } P_k \notin \Theta \\ 0 & \text{when } P_k \in \Theta \end{cases} \tag{6}$$

2.　Tasks are only executed by teachers who have the appropriate competence:

$$x_{k,i}^{\theta} \leq q_i \times g_{k,i}^{\theta}, \text{ for } k = 1 \dots m; i = 1 \dots n; \theta \in U_{\omega} \tag{7}$$

3. All tasks, $q_i$, from course $Z_i$ should be executed:

$$\left(\sum_{k=1}^{m} x_{k,i}^{\theta} = q_i\right) \Leftrightarrow \left(c_{1,i}^{\theta} = 1\right), \text{ for } i = 1 \dots n; \theta \in U_{\omega} \tag{8}$$

4. Workload of teacher $P_k$ is equal to or greater than the minimum number of working hours, $s_k$:

$$\left(\sum_{i=1}^{n} x_{k,i}^{\theta} \times l_i \geq s_k\right) \Leftrightarrow (c_{2,k}^{\theta} = 1), \text{ for } P_k \in \mathcal{P} \backslash \theta; \theta \in U_{\omega} \tag{9}$$

5. Workload of teacher $P_k$ is not greater than the maximum number of working hours, $z_k$:

$$\left(\sum_{i=1}^{n} x_{k,i}^{\theta} \times l_i \leq z_k\right) \Leftrightarrow \left(c_{3,k}^{\theta} = 1\right), \text{ for } P_k \in \mathcal{P} \backslash \theta; \theta \in U_{\omega} \tag{10}$$

6. Robustness, $R_{\mathcal{P}}^{Z}(\omega)$, is calculated as a ratio of the number of absence scenarios $|LP_{\omega}|$ for which the competency framework is robust to the absence of $\omega$ teachers to all possible disruption scenarios ($|U_{\omega}|$):

$$R_{\mathcal{P}}^{Z}(\omega) = \frac{|LP_{\omega}|}{|U_{\omega}|} \geq {}^{*}R_{\mathcal{P}}^{Z}(\omega) , \tag{11}$$

$$|LP_{\omega}| = \sum_{\theta \in U_{\omega}} c^{\theta} , \tag{12}$$

$$c^{\theta} = \prod_{i=1}^{n} c_{1,i}^{\theta} \prod_{k=1}^{m} c_{2,k}^{\theta} \prod_{k=1}^{m} c_{3,k}^{\theta} . \tag{13}$$

To summarise, the model proposed above comprises a set of decision variables (describing competency framework, the measure of its robustness and teacher allocation), discrete domains of decision variables and a set of constraints (relationships connecting the decision variables) which specify the requirements for the competency framework and the execution of planned courses.

The concepts of the competency framework and teacher allocation, $X$, are represented by decision variables $G$, $G^{\theta}$ and $X^{\theta}$. Allocation, $X^{\theta}$, in the event of teacher absence defined in set $\Theta$, which meets constraints (6)–(13), is hereafter referred to as a permissible allocation. Given the assumptions presented above, the model is deterministic.

From how the model is specified, that is being limited to defining the decision variables, variable domains and the constraints on subsets of variables, the problem under consideration belongs to the class of Constraint Satisfaction Problems (CSP).

### 3.3. Method

The structure of the proposed model that includes a set of decision variables and a set of constraints that relate those variables to one another in a natural way allows to formulate the problem in hand as a CSP and implement it in a constraint programming environment:

$$CS(\omega) = \big((\mathcal{V}(\omega), \mathcal{D}(\omega)), \mathcal{C}(\omega)\big), \tag{14}$$

where:

$\mathcal{V}(\omega)$ $= \{R_{\mathcal{P}}^{Z}(\omega), G, G^{\theta}, X^{\theta} \mid \theta \in U_{\omega}\}$—a set of decision variables which includes: robustness of competency framework $R_{\mathcal{P}}^{Z}(\omega)$, competency framework $G$, competency frameworks $G^{\theta}$ for cases when the teachers from set $\Theta$ are absent, corresponding task allocations $X^{\theta}$.

$\mathcal{D}(\omega)$ —a finite set of decision variable domains $\{R_{\mathcal{P}}^{Z}(\omega), G, G^{\theta}, X^{\theta} \mid \theta \in U_{\omega}\}$

$\mathcal{C}(\omega)$ —a set of constraints specifying the relationships between the competency framework and its robustness (constraints 1–10).

To solve $CS(\omega)$ (14), it is enough to find the values of decision variables $G$ (competency framework), $X^{\theta}$ (teacher allocation) and $R_{\mathcal{P}}^{Z}(\omega)$ (robustness of a competency framework), determined by domains $\mathcal{D}(\omega)$ for which all the constraints of set $\mathcal{C}(\omega)$ are satisfied. In other words,

what is sought is a solution that guarantees a given level of robustness, $R_{\mathcal{P}}^{Z}(\omega)$, in the case of simultaneous absences of $\omega$ teachers. In general, a CSP defined in this way can be treated as an optimisation problem. In such cases, the investigation focuses on determining the minimum competency framework, $G_{OPT}$ (e.g., one that meets the criterion of the minimum number of competence changes). In the general case, a CS (14) can be treated as an optimisation problem whose goal is to determine the minimum competency framework, $G_{OPT}$ (e.g., one that requires a minimum number of changes to be made to the baseline competency framework). CSP converted into a Constraint Optimisation Problem (COP) is given by the formula:

$$COP(\omega) = \big((\mathcal{V}(\omega), \mathcal{D}(\omega)), \mathcal{C}(\omega), F(G)\big) \tag{15}$$

where: $\mathcal{V}(\omega), \mathcal{D}(\omega)$ and $\mathcal{C}(\omega)$ are defined as in (11), and $F(G)$ is the objective function:

$$F(G) = \sum_{k=1\ldots m}^{i=1\ldots n} g_{k,i} \tag{16}$$

To solve $COP(\omega)$ (15), determine such values of decision variable $G_{OPT}$ for which all constraints given in the set $\mathcal{C}(\omega)$ are satisfied and for which function $F(G)$ has a minimum value (a minimum number of changes have to be made to the original competency framework, $G$) or, to put it differently, returns a minimum competency framework. In general, $COP(\omega)$ (15) synthesises minimal competency frameworks robust to simultaneous absences of $\omega$ teachers.

The model of the synthesis problem $COP(\omega)$ presented in Figure 2 illustrates the procedure of finding a competency framework, $G$, with a given level of robustness ($R_{\mathcal{P}}^{Z}(\omega) \geq {}^{*}R_{\mathcal{P}}^{Z}(\omega)$) for the case of individual teacher absence ($\omega = 1$). A specific level of robustness can be obtained from the introduction of decision variables $G^{\{P_1\}}, G^{\{P_2\}}, \ldots, G^{\{P_m\}}$, which represent the competency frameworks for the cases of individual teacher absences: $U_1 = \{\{P_1\}, \{P_2\}, \ldots, \{P_m\}\}$. Full robustness ($R_{\mathcal{P}}^{Z}(\omega) = 1$) is reached when there exists a framework, $G$, for which each $G^{\Theta}$ guarantees teacher allocation $X^{\Theta}$ that meets constraints (6)–(13) ($c^{\Theta} = 1$). In other words, the solution to problem $COP(\omega)$ (15) is a minimal competency framework, $G_{OPT}$, that guarantees completion of all tasks for all cases of single-teacher absences.

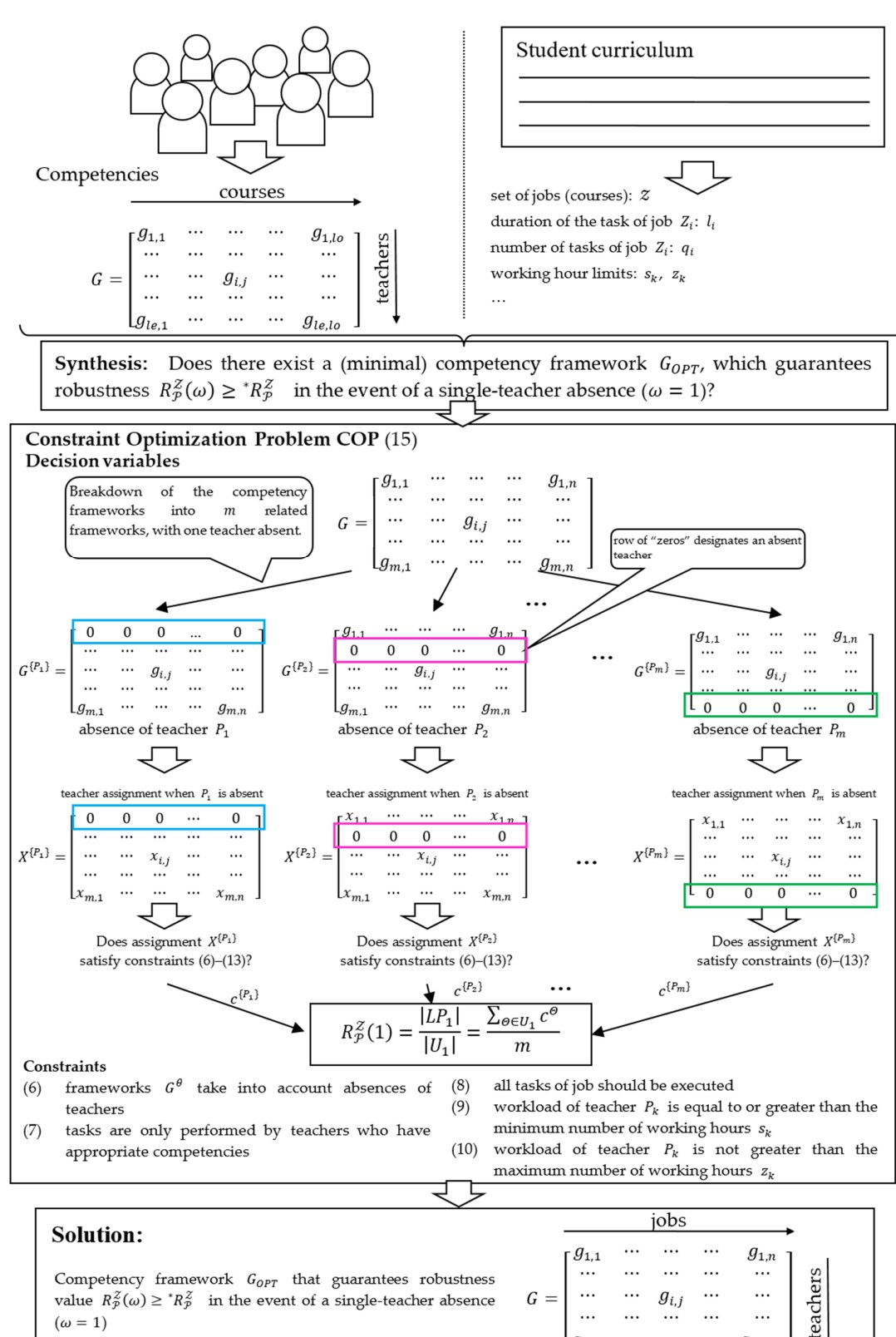

**Figure 2.** Model of Constraint Optimisation Problem (COP)-based synthesis of a robust competency framework.

The method shown in Figure 3 was developed for the synthesis of a minimal competency framework robust to disruptions resulting from various factors, including simultaneous absences of $\omega$ teachers or loss of qualifications (competencies). Decision-makers are aware of the possible occurrence of a specific set of disruptions. Our decision support tool is designed to answer the question regarding the analysis (evaluation) of the robustness of the competency framework to a selected set of disruptions. When the answer is positive (the competency framework is sufficiently flexible to allow the available personnel to complete all courses), the university can proceed to execute the tasks without fear of disruption. When the answer is negative (the competency framework is insufficiently flexible to allow the available teachers to complete all courses), the decision-maker may then implement the method proposed in this paper to look for an answer to the question regarding the synthesis of the competency framework (solution to $COP(\omega)$ (15)), i.e., to search available data (in this case, the competency framework) to find a teacher allocation that meets specific expectations (e.g., robustness of the competency framework to the selected set of disruptions). When the answer is positive, the decision-maker obtains a set of alternative competency frameworks that guarantee the organisation's robustness to a selected set of disruptions. It is on the basis of this set of admissible alternative frameworks, that substantiated decisions regarding issues such as further development of the staff can be made. It is the decision-maker's responsibility to choose the most favourable variant (one that meets a criterion of the choice). A negative answer informs the decision-maker that it is impossible in the given organisation to build a competency framework robust to the selected set of disruptions and consider changing (increasing) working time limits, employing new staff, outsourcing to temporary workers, etc.

Implemented in LINGO programming environments with a hybrid approach proposed by Wikarek and Sitek [62], the method was used in a series of computational experiments testing the synthesis competency framework of teaching staff in a real university environment. Preliminary results have confirmed the benefits of the adopted declarative approach that allow designing an open structure model, as well as an implementation of the method derived from this model and used in interactive decision support systems (DSS) dedicated to online solving of staff planning problems [63].

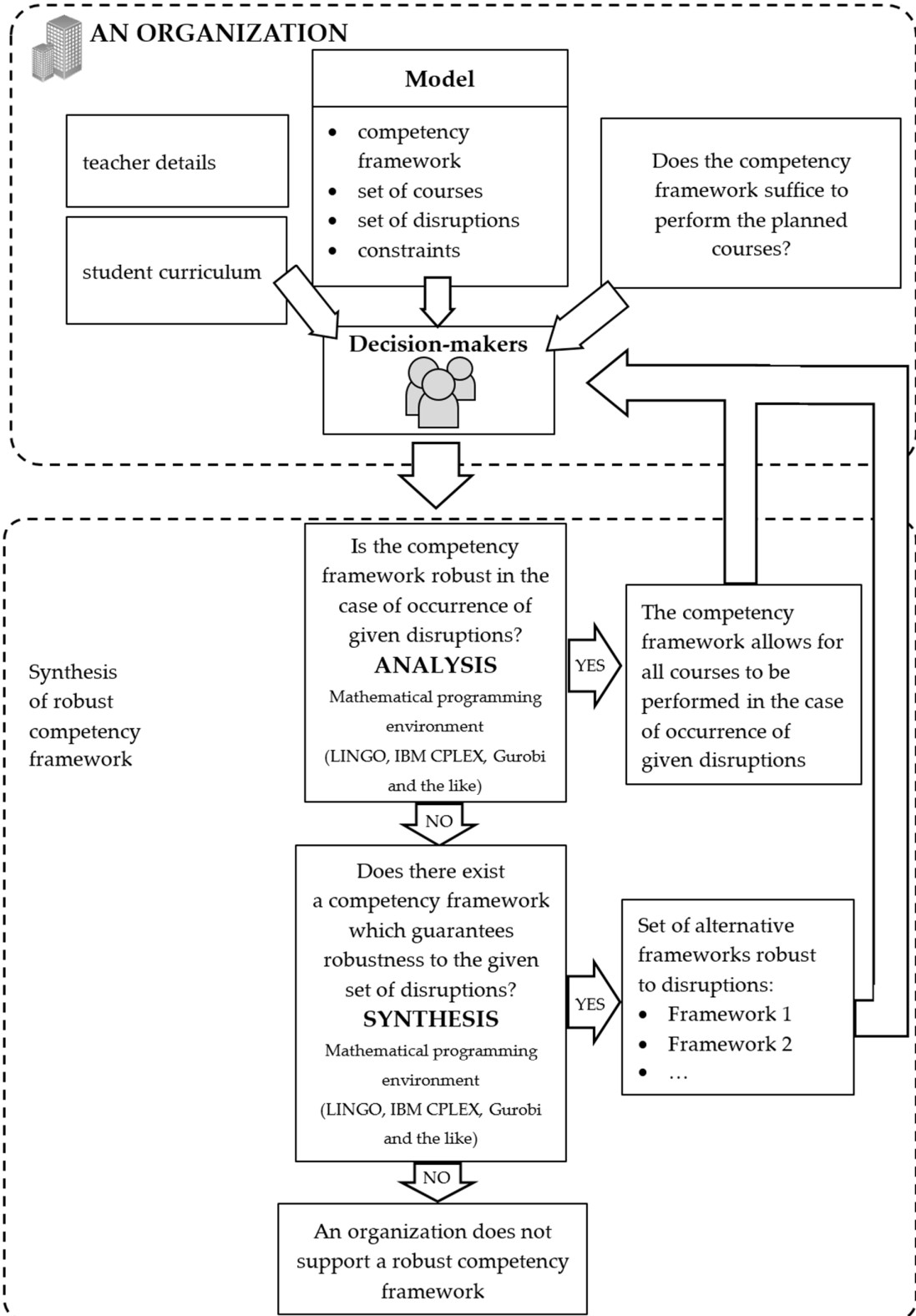

**Figure 3.** The synthesis method of a competency framework robust to the selected set of disruptions.

## 4. University Staff Allocation Planning—Case of Course Cast

The effectiveness of the aforementioned method has been verified on the real data collected in the process of teaching at the Koszalin University of Technology in the academic year 2019/2020. The university is a public institution of higher education in Poland and carries out the educational activities and scientific research in disciplines related primarily to the directions of development of the Middle Pomerania region. Of six faculties of this university, the Faculty of Electronics and Computer Science (FECS) was selected for further consideration.

FECS curriculum: In the academic year 2019/2020, two fields of study were conducted at the faculty that consisted of $n$ = 214 courses: $Z = \{Z_1, Z_2, ..., Z_{214}\}$ (including full-time and part-time BSc, MSc and PhD courses), with a total of 14,100 h. The number $q_i$ and duration $l_i$ of tasks of course $Z_i$ derived from the faculty syllabus are collected in Table 5.

**Table 5.** Faculty of Electronics and Computer Science (FECS) syllabus.

| $Z_i$ | $q_i$ | $l_i$ |
|---|---|---|
| $Z_1$: History of technics 1 | 16 | 5 |
| $Z_2$: History of technics 2 | 5 | 5 |
| $Z_3$: Inventics | 12 | 5 |
| $Z_4$: Economics | 9 | 5 |
| $Z_5$: Foundations of mathematical analysis | 20 | 5 |
| … | … | … |
| $Z_{74}$: Programming in. NET environment | 21 | 5 |
| … | … | … |
| $Z_{213}$: Distributed information processing systems | 6 | 5 |
| $Z_{214}$: Artificial intelligence methods | 6 | 5 |

FECS teaching staff: The classes are given by a team of $m$ = 49 teachers $\mathcal{P} = \{P_1, P_2, ..., P_{49}\}$. The competency framework, $G$, established from the survey results, shows which teacher can teach what classes:

- May conduct: $g_{k,i} = 1$,
- May conduct if it gains the missing competencies: $g_{k,i} \in \{0,1\}$,
- May not conduct and cannot get the missing competencies: $g_{k,i} = 0$.

Part of the considered competency framework, $G$, is illustrated in Table 6. Following the provisions of the General Data Protection Regulation, the data were pseudonymised.

In addition, the lower ($s_k$) and upper ($z_k$) limit of hours to be implemented for each teacher is assumed (Table 7). For example, $P_1$: Mills has $s_k$ = 180 h and $z_k$ = 360 h, $P_2$: Garner has $s_k$ = 360 h and $z_k$ = 600 h, and so on. The limits are unchangeable for each teacher.

FECS teacher allocation: Teacher allocation, $X$, valid in the academic year 2019/2020, is presented in Table 8. The considered teacher allocation, $X$, follows these requirements:

- Tasks from course $Z_i$ can only be executed by a competent teacher,
- Teacher working time limits ($s_k$ and $z_k$) may not be exceeded.

In other words, accepted teacher allocation, $X$, is sufficient for the given FECS set of courses.

The data allowed us to verify the developed method and to determine, at a later stage, $G$ competency frameworks absorbing the FECS disruptions, in particular:

1. Method verification based on historical data: The verification includes an assessment of whether the use of the developed method would lead to the determination of $G$, a competency framework that would secure FECS (robustness $R_{\mathcal{P}}^Z(\omega) = 1$) against the effects inferred from the teacher absence in the academic year 2019/2020, e.g., the need to hire an additional teacher. In the case of successful assessment:
2. Use of the developed method aimed at synthesis of competency frameworks robust to selected kinds of disruptions including:

- Absence of $\omega = 1, \dots, 3$ teachers,
- Absence of teachers from the age group of pre-retirement and retirement employees.

**Table 6.** Competency framework, $G$, of FECS teaching staff (source: https://github.com/erykszw/TAP).

| $G$ | $Z_1$: History of Technics 1 | $Z_2$: History of Technics 2 | $Z_3$: Inventics | $Z_4$: Economics | $Z_5$: Foundations of mathematical | $Z_6$: Mathematical Analysis and Linear Algebra | … | … | $Z_{74}$: Programming in .NET | … | … | $Z_{212}$: Selected Branches of Physics | $Z_{213}$: Distributed Information | $Z_{214}$: Artificial Intelligence |
|---|---|---|---|---|---|---|---|---|---|---|---|---|---|---|
| $P_1$: Mills | 1 | 1 | 0 | 0 | 0 | 0 | … | … | 0 | … | … | 1 | 0 | 0 |
| $P_2$: Garner | 0 | 0 | 0 | 0 | 0 | 0 | … | … | 0 | … | … | 0 | 0 | 0 |
| $P_3$: Ray | 0 | 0 | 0 | 0 | 0 | 0 | … | … | 0 | … | … | 0 | 0 | 0 |
| $P_4$: MacPherson | 0 | 0 | 0 | 0 | 0 | 0 | … | … | {0,1} | … | … | 0 | 0 | 0 |
| $P_5$: Burnham | 0 | 0 | 0 | 0 | 0 | 0 | … | … | {0,1} | … | … | 0 | 0 | 0 |
| $P_6$: Davis | 0 | 0 | 0 | 0 | 0 | 0 | … | … | 0 | … | … | 0 | 0 | {0,1} |
| $P_7$: Crockett | {0,1} | {0,1} | {0,1} | 0 | 0 | 0 | … | … | {0,1} | … | … | 0 | 1 | {0,1} |
| $P_8$: Hudson | {0,1} | {0,1} | {0,1} | 0 | 0 | 0 | … | … | 0 | … | … | 0 | 0 | 0 |
| … | … | … | … | … | … | … | … | … | … | … | … | … | … | … |
| $P_{18}$: Roach | 0 | 0 | 0 | 0 | 0 | 0 | … | … | 1 | … | … | 0 | 0 | {0,1} |
| … | … | … | … | … | … | … | … | … | … | … | … | … | … | … |
| $P_{47}$: Fox | 0 | 0 | 0 | 1 | 0 | 0 | … | … | 0 | … | … | 0 | 0 | 0 |
| $P_{48}$: Porterfield | 0 | 0 | 0 | 0 | 0 | 0 | … | … | 0 | … | … | 0 | 0 | 0 |
| $P_{49}$: Johnson | 0 | 0 | 0 | 0 | 0 | 0 | … | … | 0 | … | … | 0 | 0 | 0 |

**Table 7.** FECS teaching staff hour limits.

| Teacher | $s_k$ | $z_k$ | Teacher | $s_k$ | $z_k$ | Teacher | $s_k$ | $z_k$ |
|---|---|---|---|---|---|---|---|---|
| $P_1$: Mills | 180 | 360 | $P_{18}$: Roach | 180 | 360 | $P_{35}$: Morrow | 180 | 360 |
| $P_2$: Garner | 360 | 600 | $P_{19}$: Schneider | 240 | 480 | $P_{36}$: Fitch | 240 | 480 |
| $P_3$: Ray | 180 | 360 | $P_{20}$: Reyes | 240 | 480 | $P_{37}$: Clay | 240 | 480 |
| $P_4$: MacPherson | 180 | 360 | $P_{21}$: Barnes | 120 | 240 | $P_{38}$: Manning | 240 | 480 |
| $P_5$: Burnham | 360 | 600 | $P_{22}$: Meyer | 360 | 600 | $P_{39}$: Ramsey | 180 | 360 |
| $P_6$: Davis | 120 | 240 | $P_{23}$: Sharpe | 120 | 240 | $P_{40}$: Hansen | 240 | 480 |
| $P_7$: Crockett | 128 | 360 | $P_{24}$: Sinclair | 160 | 360 | $P_{41}$: Thorpe | 180 | 360 |
| $P_8$: Hudson | 240 | 480 | $P_{25}$: Mahoney | 180 | 360 | $P_{42}$: Rice | 340 | 600 |
| $P_9$: Whittaker | 240 | 480 | $P_{26}$: Kirkland | 240 | 480 | $P_{43}$: Whitehead | 240 | 480 |
| $P_{10}$: Middleton | 360 | 600 | $P_{27}$: Slaughter | 240 | 500 | $P_{44}$: Lacroix | 20 | 120 |
| $P_{11}$: Sloan | 330 | 600 | $P_{28}$: Gardner | 360 | 600 | $P_{45}$: Nichols | 50 | 120 |
| $P_{12}$: Flynn | 360 | 600 | $P_{29}$: Richardson | 190 | 480 | $P_{46}$: Cooley | 20 | 120 |
| $P_{13}$: Pope | 240 | 480 | $P_{30}$: Byrne | 240 | 480 | $P_{47}$: Fox | 30 | 120 |
| $P_{14}$: Buckley | 240 | 480 | $P_{31}$: Curran | 240 | 480 | $P_{48}$: Porterfield | 150 | 300 |
| $P_{15}$: Johnston | 360 | 600 | $P_{32}$: Owens | 180 | 400 | $P_{49}$: Johnson | 50 | 100 |
| $P_{16}$: Bullock | 180 | 360 | $P_{33}$: Hoover | 345 | 600 | | | |
| $P_{17}$: Dowling | 360 | 600 | $P_{34}$: Reynolds | 240 | 480 | | | |

**Table 8.** Teacher allocation, *X*, of the teaching staff employed by FECS (source: https://github.com/erykszw/TAP).

| *X* | $Z_1$: History of Technics 1 | $Z_2$: History of Technics 2 | $Z_3$: Inventics | $Z_4$: Economics | $Z_5$: Foundations of mathematical Analysis | $Z_6$: Mathematical Analysis and Linear Algebra | ... | ... | $Z_{74}$: Programming in .NET Environment | ... | ... | $Z_{212}$: Selected Branches of Physics | $Z_{213}$: Distributed Information Processing Systems | $Z_{214}$: Artificial Intelligence Methods |
|---|---|---|---|---|---|---|---|---|---|---|---|---|---|---|
| $P_1$: Mills | 0 | 0 | 0 | 0 | 0 | 0 | ... | ... | 0 | ... | ... | 15 | 0 | 0 |
| $P_2$: Garner | 0 | 0 | 0 | 0 | 0 | 0 | ... | ... | 0 | ... | ... | 0 | 0 | 0 |
| $P_3$: Ray | 0 | 0 | 0 | 0 | 0 | 0 | ... | ... | 0 | ... | ... | 0 | 0 | 0 |
| $P_4$: MacPherson | 0 | 0 | 0 | 0 | 0 | 0 | ... | ... | 0 | ... | ... | 0 | 0 | 0 |
| $P_5$: Burnham | 0 | 0 | 0 | 0 | 0 | 0 | ... | ... | 0 | ... | ... | 0 | 0 | 0 |
| $P_6$: Davis | 0 | 0 | 0 | 0 | 0 | 0 | ... | ... | 0 | ... | ... | 0 | 0 | 0 |
| $P_7$: Crockett | 0 | 0 | 0 | 0 | 0 | 0 | ... | ... | 0 | ... | ... | 0 | 0 | 0 |
| ... | ... | ... | ... | ... | ... | ... | ... | ... | ... | ... | ... | ... | ... | ... |
| $P_{18}$: Roach | 0 | 0 | 0 | 0 | 0 | 0 | ... | ... | 80 | ... | ... | 0 | 0 | 0 |
| ... | ... | ... | ... | ... | ... | ... | ... | ... | ... | ... | ... | ... | ... | ... |
| $P_{47}$: Fox | 0 | 0 | 0 | 0 | 0 | 0 | ... | ... | 0 | ... | ... | 0 | 0 | 0 |
| $P_{48}$: Porterfield | 0 | 0 | 0 | 0 | 0 | 0 | ... | ... | 0 | ... | ... | 0 | 0 | 0 |
| $P_{49}$: Johnson | 0 | 0 | 0 | 0 | 0 | 0 | ... | ... | 0 | ... | ... | 0 | 0 | 0 |

*4.1. Method Verification Based on Historical Data*

During the 2019/2020 academic year, teacher $P_{18}$ was absent: Roach was in the hospital and consequently quit his job. In order to maintain the continuity of classes, it was resolved that a new competent teacher would be employed to take over courses $Z_{47}, Z_{48}, Z_{74}, Z_{106}, Z_{121}, Z_{125}, Z_{211}$ conducted thus far by teacher $P_{18}$. The necessary organisational changes incurred further costs associated with reorganising the class schedule, training a new teacher, etc.

The question that naturally arises is: Could this type of situation be avoided? Or alternatively speaking: Could it have been previously possible to supplement the competencies of a selected teacher (or teachers) so that they could replace an absent colleague? The method presented in this work is dedicated to resolving such questions. In order to illustrate its applicability, let us consider datasets collected in Tables 6–8, for which competency framework $G_{OPT}$ robustness ($R_P^Z(\omega) = 1$) to teacher $P_{18}$: Roach's absence is sought. In other words, the question is:

*Does there exist a minimal competency framework $G_{OPT}$ of FECS that guarantees robustness value $R_P^Z(\omega) = 1$ in the event of absence of teacher $P_{18}$: Roach?*

The method has been implemented in the LINGO environment (Intel i7-4770, 8GB RAM). The optimal solution, $G_{OPT}$ (source: GitHub), was obtained in under 1 s. In competency framework $G_{OPT}$, there is an acceptable teacher allocation, *X*, allowing other teachers $P_{29}$, $P_{41}$, $P_{44}$ and $P_7$ to conduct courses $Z_{47}, Z_{48}, Z_{74}, Z_{106}, Z_{121}, Z_{211}$ and $Z_{125}$, which were initially assigned to the absent teacher $P_{18}$:

- Teacher $P_{29}$: Richardson can take over the substitution by conducting classes in:

    - Course $Z_{47}$ (50 h, 10 tasks)
    - Course $Z_{48}$ (75 h, 15 tasks)
    - Course $Z_{74}$ (80 h, 16 tasks)

- Teacher $P_{41}$: Thorpe can take over the substitution by conducting classes in:
    - Course $Z_{106}$ (15 h, 3 tasks)

- Course $Z_{211}$ (20 h, 4 tasks)

- Teacher $P_{44}$: Lacroix can take over the substitution by conducting classes in:

  - Course $Z_{121}$ (30 h, 6 tasks)

- Teacher $P_7$: Crockett can take over the substitution by conducting classes in:

  - Course $Z_{125}$ (75 h, 15 tasks)

In the new competency framework, $G_{OPT}$, teacher $P_7$ is assumed to take over a single competency in conducting classes from course $Z_{125}$. The $G_{OPT}$ competency framework ensures the continuity of classes in the absence of teacher $P_{18}$: Roach without having to employ an additional replacement tutor.

The illustrated option of supplementing competencies in the scope of conducting classes from course $Z_{125}$ by teacher $P_7$ is one of several solutions obtained, others being, for example:

- Teacher $P_{22}$: Meyer complements the competency from course $Z_{125}$,
- Teacher $P_{43}$: Whitehead complements the competency from course $Z_{125}$, etc.

The solutions presented above include cases requiring filling in for competencies in only one course. In practice, it emerges that a sufficiently early supplementation of competencies by one teacher would protect FECS against the effects of resignation from the work of teacher $P_{18}$: Roach.

In general, employee absences are usually unexpected (unplanned), which curbs predicting which teachers and in what number will be absent. Therefore, to provide the framework with suitable absorption qualities, absences of several teachers must be simulated. Examples of the synthesis of the FECS competency framework guaranteeing robustness to this type of interference are presented in the next section.

### 4.2. Synthesis of Competency Framework Robust to a Simultaneous Absence of $\omega$ Teachers

In the previous experiment, the $G_{OPT}$ competency framework was determined to protect FECS against the effects (necessity of additional employment) of teacher $P_{18}$: Roach's absence. In the next stage, the $G_{OPT}$ competency framework was synthesised to obtain $R_{\mathcal{P}}^{Z}(\omega) = 1$ robustness to absenteeism of: (a) any single teacher ($\omega = 1$), (b) any two teachers ($\omega = 2$) and (c) any three teachers ($\omega = 3$). In experiments, the FECS data were derived from Tables 6–8. For such data, the emerging question is:

*Does there exist a (minimal) competency framework, $G_{OPT}$, which guarantees robustness value $R_{\mathcal{P}}^{Z}(\omega) = 1$ in in the event of absence of $\omega$ teachers ($\omega = 1,2,3$)?*

As in the previous case, the method was fed in the LINGO environment (Intel i7-4770, 8GB RAM) and the solutions were calculated in 14.2 s (option of $\omega = 1$), 49.4 s (option of $\omega = 2$) and 1185 s (option of $\omega = 3$). Consequently, the answer to this question is negative, i.e., there is no such form among the permissible competency framework that guarantees the maximum robustness level $R_{\mathcal{P}}^{Z}(\omega) = 1$, however, for disruptions in $\omega = 1,2,3$ cases, the corresponding levels are:

(a) $R_{\mathcal{P}}^{Z}(1) = 0.77$,
(b) $R_{\mathcal{P}}^{Z}(2) = 0.58$,
(c) $R_{\mathcal{P}}^{Z}(3) = 0.43$.

Competency frameworks $G_{OPT}^1$, $G_{OPT}^2$ and $G_{OPT}^3$ (where the superscript refers to variants $\omega = 1,2,3$) respectively, guaranteeing the above values of robustness $R_{\mathcal{P}}^{Z}(\omega)$ are presented in GitHub.

The results above describe that the current competencies of the staff members allow for the development of teacher competencies in accordance with the designated framework, $G_{OPT}^1$ ($R_{\mathcal{P}}^{Z}(1) = 0.77$), to provide the faculty with the capacity to absorb the effects of 77% possible scenarios of an absence of a single teacher. Nevertheless, it is shown that further development of competencies will not improve the robustness of the competency framework, whose maximum value is equal to 0.77.

In other words, there are no such changes in the current competency framework, $G$ (see Table 6), that will guarantee robustness to the absence of any single teacher at the level $R_{\mathcal{P}}^Z(1) > 0.77$.

Due to the incapability of the method to produce a solution to the synthesis problem, the possibility of increasing the teaching staff numbers should be considered, that is:

*Teachers with what competencies should be employed to obtain $G$ competency framework whose robustness level, $R_{\mathcal{P}}^Z(\omega) = 1$, corresponds to situations when $\omega$ teachers are absent ($\omega = 1,2,3$)?*

The method determines acceptable solutions in 14.4 s (option of $\omega = 1$), 51.5 s (option of $\omega = 2$) and 1131 s (option of $\omega = 3$). The calculations showed that in order to obtain a competency framework, $G$, guaranteeing robustness level $R_{\mathcal{P}}^Z(\omega) = 1$, respectively for $\omega = 1,2,3$, additional teachers with suitable competencies should be employed. The results of the calculations ($G_{OPT}^1$, $G_{OPT}^2$, $G_{OPT}^3$—see GitHub) showed that in order to obtain full robustness absorption, ($R_{\mathcal{P}}^Z(\omega) = 1$ for $\omega = 1,2,3$), a team of teachers must possess the following competencies in:

(a) Conducting 21 courses: $Z_4, Z_7, Z_{23}, Z_{24}, Z_{25}, Z_{45}, Z_{93}, Z_{103}, Z_{114}, Z_{131}, Z_{132}, Z_{134}, Z_{135}, Z_{157}, Z_{158},$
    $Z_{166}, Z_{168}, Z_{169}$ and $Z_{170}$ ($\omega = 1$).
(b) Conducting 71 courses: $Z_4, Z_5, Z_6, Z_7, Z_9, Z_{10}, Z_{19}, Z_{21}, Z_{22}, Z_{23}, Z_{24}, Z_{25}, Z_{26}, Z_{27}, Z_{28},$
    $Z_{29}, Z_{30}, Z_{34}, Z_{45}, Z_{51}, Z_{55}, Z_{56}, Z_{58}, Z_{77}, Z_{78}, Z_{79}, Z_{84}, Z_{86}, Z_{93}, Z_{99}, Z_{101}, Z_{102}, Z_{103}, Z_{104}, Z_{107}, Z_{111},$
    $Z_{114}, Z_{115}, Z_{117}, Z_{120}, Z_{130}, Z_{131}, Z_{132}, Z_{133}, Z_{134}, Z_{135}, Z_{136}, Z_{137}, Z_{149}, Z_{153}, Z_{156}, Z_{157}, Z_{158}, Z_{159}, Z_{160},$
    $Z_{161}, Z_{162}, Z_{164}, Z_{165}, Z_{166}, Z_{168}, Z_{169}, Z_{170}, Z_{179}, Z_{191}, Z_{196}, Z_{201}, Z_{203}, Z_{208}$ and $Z_{212}$ ($\omega = 2$).
(c) Conducting 129 courses ($\omega = 3$): $G_{OPT}^3$, see in GitHub.

The obtained results show the competencies that newly recruited teachers should have in order for the $G_{OPT}$ competency framework to be fully robust, ($R_{\mathcal{P}}^Z(\omega) = 1$), to the disruptions discussed in this subsection. However, the number of teachers to be employed has not been computed. How many teachers should be hired depends on available candidates and their competencies. Selecting the team of teachers with a specific set of competencies is the next stage of research, extending the developed model by elements related to the framework of the newly employed academic staff.

*4.3. Synthesis of Competency Framework Robust to Absenteeism Caused by Pre-Retirement Aged Teachers*

As part of the conducted series of experiments, another one concerned the synthesis of the FECS competency framework robust to the absence of a selected group of teachers, for instance, pre-retirement aged teachers, who are at risk of retiring at any time. There are 9 such teachers in FECS:

$$EM = \{P_1: \text{Mills}, P_3: \text{Ray}, P_7: \text{Crockett}, P_{16}: \text{Bullock}, P_{18}: \text{Roach}, P_{21}: \text{Barnes},$$
$$P_{24}: \text{Sinclair}, P_{39}: \text{Ramsey}, P_{41}: \text{Thorpe}\}$$

For such a set, $EM$, the answer to the following question is sought:

*Does there exist a (minimal) competency framework, $G_{OPT}$, of FECS that guarantees robustness value $R_{\mathcal{P}}^Z(\omega) = 1$ in the event of absence of $\omega$ teachers from set $EM$ ($\omega = 1, \dots, 9$)?*

A positive answer was obtained (calculation time 2.2 s) only for the variant of an absence of a single teacher ($\omega = 1$). The resulting $G_{OPT}^1$ competency framework, which has been placed in GitHub, shows that there exists a set of teachers $\{P_3, P_8, P_{22}, P_{31}, P_{33}, P_{36}, P_{38}, P_{41}, P_{42}, P_{43}, P_{46}\}$ able to broaden their competencies and consequently enabling their allocation, $X$, that will ensure the continuity of classes without having to hire new staff.

For other variants of the disruption under consideration ($\omega = 2, \dots, 9$), it is not possible to obtain a $G_{OPT}$ competency framework that would guarantee the full robustness level $R_{\mathcal{P}}^Z(\omega) = 1$. In other words, the available teaching staff is not able to secure the FECS education process in the absence of more than one $EM$ teacher. The calculations established that the maximum level of robustness for $\omega = 2, \dots, 9$, is respectively: $R_{\mathcal{P}}^Z(2) = 0.92$; $R_{\mathcal{P}}^Z(3) = 0.76$; $R_{\mathcal{P}}^Z(4) = 0.55$; $R_{\mathcal{P}}^Z(5) = 0.31$; $R_{\mathcal{P}}^Z(6) = 0.11$; $R_{\mathcal{P}}^Z(7) = 0$; $R_{\mathcal{P}}^Z(8) = 0$, $R_{\mathcal{P}}^Z(9) = 0$.

Note that the maximum robustness value for $\omega = 7, \dots, 9$ is equal to 0. This means that in extreme situations, such as the resignation of 7 (or more) pre-retirement aged teachers, it is not

possible to maintain the continuity of classes without employing additional teaching staff. Consequently, assuming an increase in the number of newly employed teachers, an attempt to synthesise a proper competency framework was made. In this context, it means seeking the answer to the following question:

*Teachers with what competencies should be employed to obtain $G$ competency framework whose robustness level $R_{\mathcal{P}}^{Z}(\omega) = 1$ corresponds to the event of absence of $\omega$ teachers from set $EM$ $(\omega = 2, \ldots, 9)$?*

The results of the calculations (obtained in 39 s) showed that achieving the full robustness level, $R_{\mathcal{P}}^{Z}(\omega) = 1$ for $\omega = 2$, is conditioned by employing teachers with competencies in 7 courses: $Z_9, Z_{10}, Z_{136}, Z_{137}, Z_{179}, Z_{208}$ and $Z_{212}$, as well as for $\omega = 3, \ldots, 9$, is conditioned by employing teachers with competencies in 9 courses:: $Z_9, Z_{10}, Z_{130}, Z_{136}, Z_{137}, Z_{146}, Z_{179}, Z_{208}$ and $Z_{212}$.

*4.4. Quantitative Experiments*

In addition to the computations presented in the preceding sections, we attempted to illustrate selected applications of the developed method and to verify our approach in several quantitative experiments of scalability. With this in mind, the effectiveness of the method in solving competency framework analysis and synthesis problems subject to the occurrence of disruptions of a various scale was assessed. Using data collected in Tables 6–8, solutions were sought to guarantee levels of robustness, $R_{\mathcal{P}}^{Z}(\omega) \geq 0.2.1$, in the absence of $\omega = 1, \ldots, 7$ teachers. The results from the experiments, carried out in the LINGO software environment, are presented in Tables 9–11.

**Table 9.** Analysis of robustness, $R_{\mathcal{P}}^{Z}(\omega)$, of FECS competency framework (Table 6).

| Number of Absent Teachers, $\omega$ | Robustness Level, $R_{\mathcal{P}}^{Z}(\omega)$ | Calculation Time (s) |
|---|---|---|
| 1 | 0.35 | 0.9 |
| 2 | 0.1 | 1.1 |
| 3 | 0.03 | 1.4 |
| 4 | 0.01 | 2.2 |
| 5 | 0 | 4.5 |
| 6 | 0 | 6.8 |
| 7 | 0 | 9.2 |

Clearly, in the case of the competency framework analysis problem (Table 9), the answer can be obtained in an online mode (1200 s) even when $\omega = 7$. In turn, in the cases of the competency framework synthesis problems (Tables 10 and 11), the online decision support is limited to $\omega = 3$ (i.e., synthesis of a robust competency framework for a 3-teacher absence)—computation times exceeded the assumed 1200 s.

Note that the presented experiments determine the maximum capabilities of the available FECS staff. The results indicate that in the case of a simultaneous absence of 5 (or more) teachers (specified by the competency framework, $G$—see Table 6), the remaining teachers are unable to ensure the continuity of FECS classes, regardless of the competency set. Protection against these types of cases is possible provided that additional staff are employed to fill in for 184 courses.

Experiments carried out for larger-scale objects (Table 12: $m = 50 \ldots 200$, $n = 300 \ldots 600$) show that the proposed approach can be used to support decisions on the synthesis of robust competency frameworks for instances of no more than 150 teachers and 500 courses.

The results from the experiments demonstrate that the proposed approach can be implemented in Decision Support Systems (DSS) used in online task assignment. In this context, it seems that the approach may prove particularly powerful in solving teacher allocation problems of AAPDA and AAPDO types. As a result of these investigations, suggestions were identified for future research to develop a computational module that could be applied as a software overlay for commercially available decision-support systems for human resources management applications.

**Table 10.** Synthesis of competency framework, $G_{OPT}$, following robustness level condition $R_{\mathcal{P}}^{Z}(\omega) \geq {}^{*}R_{\mathcal{P}}^{Z}(\omega)$ assumed on $\omega$ teacher absenteeism.

| Number of Absent Teachers, $\omega$ | Expected Robustness Level, ${}^{*}R_{\mathcal{P}}^{Z}(\omega)$ | Obtained Robustness Level, $R_{\mathcal{P}}^{Z}(\omega)$ | Number of Changes Introduced to the Competency Framework, $G$ | Calculation Time (s) |
|---|---|---|---|---|
| | 0.2 | 0.23 | 9 | 13.1 |
| | 0.4 | 0.49 | 62 | 13.5 |
| 1 | 0.6 | 0.77 | 138 | 14.2 |
| | 0.8 | [1)]✖ | ✖ | 14.5 |
| | 1 | ✖ | ✖ | 14.9 |
| | 0.2 | 0.29 | 121 | 47.7 |
| | 0.4 | 0.58 | 415 | 49.4 |
| 2 | 0.6 | ✖ | ✖ | 51.6 |
| | 0.8 | ✖ | ✖ | 52.8 |
| | 1 | ✖ | ✖ | 54.4 |
| | 0.2 | 0.27 | 170 | 1068 |
| | 0.4 | 0.43 | 660 | 1185 |
| 3 | 0.6 | ✖ | ✖ | >1200 |
| | 0.8 | ✖ | ✖ | >1200 |
| | 1 | ✖ | ✖ | >1200 |
| | 0.2 | 0.31 | 752 | >1200 |
| | 0.4 | ✖ | ✖ | >1200 |
| 4 | 0.6 | ✖ | ✖ | >1200 |
| | 0.8 | ✖ | ✖ | >1200 |
| | 1 | ✖ | ✖ | >1200 |

[1)] ✖—no acceptable solution, i.e., there is no competency framework, *G*, which guarantees an expected value of robustness: $R_{\mathcal{P}}^{Z}(\omega) \geq {}^{*}R_{\mathcal{P}}^{Z}(\omega)$.

**Table 11.** Synthesis of competency framework, $G_{OPT}$, under robustness level condition $R_{\mathcal{P}}^{Z}(\omega) = 1$ assumed on $\omega$ teacher absenteeism while taking into account the employment of additional staff.

| Number of Absent Teachers, $\omega$ | Obtained Robustness Level, $R_{\mathcal{P}}^{Z}(\omega)$ | Number of Competencies in the Team of Newly Employed Teachers | Calculation Time (s) |
|---|---|---|---|
| 1 | 1 | 21 | 14.4 |
| 2 | 1 | 71 | 51.5 |
| 3 | 1 | 129 | 1131 |
| 4 | 1 | 155 | >1200 |
| 5 | 1 | 184 | >1200 |
| 6 | 1 | 197 | >1200 |
| 7 | 1 | 204 | >1200 |

**Table 12.** Synthesis of competency framework, $G_{OPT}$, guaranteeing full robustness ($R_{\mathcal{P}}^{Z}(\omega) = 1$) for different numbers of teachers and courses.

| Number of Teachers, $m$ | Number of Courses, $n$ | Number of Absent Teachers, $\omega$ | Calculation Time (s) |
|---|---|---|---|
| 50 | 300 | 1 | 25 |
| 50 | 300 | 2 | 53 |
| 50 | 300 | 3 | 1005 |
| 100 | 400 | 1 | 134 |
| 100 | 400 | 2 | 345 |
| 100 | 400 | 3 | 6200 |
| 150 | 500 | 1 | 234 |
| 150 | 500 | 2 | 865 |
| 150 | 500 | 3 | 11,240 |
| 200 | 600 | 1 | 1540 |
| 200 | 600 | 2 | 5980 |
| 200 | 600 | 3 | >18,000 |

## 5. Conclusions

The project was undertaken to design a method for synthesising competency frameworks robust to selected sets of disruptions in the context of planning the available academic staff allocation (teachers with specific competencies) to curriculum courses (that require specific teacher competencies). The factor that our solution was expected to provide robustness against was employee absence. The tool is useful in determining additional (redundant) competencies that organisations need to possess in order to compensate for the competencies lost as a result of employee absenteeism. The measure of robustness level, $R_{\mathcal{P}}^{Z}(\omega)$, incorporated in this method enables the analysis (checking the robustness of the adopted allocation) and synthesis (selecting of appropriate competencies) of competency frameworks, guaranteeing continuity of the ongoing education process in the absence of teachers. The introduced measure is part of the declarative model implemented in the authors' method of interactive support for the synthesis of robust competency frameworks.

Although our method concerned employee absenteeism, it could be, in fact, applied to organisations experiencing other types of singular disruption, be it handling of additional, previously unplanned orders or changes in the duration of activities. In this context, the future work will explore the model capabilities to handle other disruptions that are commonly encountered in everyday practice. An important limitation of the adopted reference model is its discreet nature. In reality, the data are often of an uncertain, stochastic or fuzzy character (e.g., an approximately 5-day duration of an activity), which implies the need to take into account an appropriate, stochastic or fuzzy representation.

The experimental part of the study allowed us to verify the proposed method based on the actual historical data from a technical university (Faculty of Electronics and Computer Science of Koszalin University of Technology). The effectiveness of the method has been confirmed in solving large-scale problems, however, not exceeding 150 teachers and 500 courses, which confirms its applicability in small- and medium-sized organisations.

Compared to existing methods [35,48–50,52–57], the developed reference model copes with two types of inherent issues, that is the competency framework robustness analysis and the problem of competency framework synthesis robust to selected types of disruptions. The adopted paradigm of declarative modelling, unlike the currently available mathematical programming methods, allows direct implementation of the developed model in commercially available programming platforms (e.g., LINGO, IBM ILOG CPLEX, Gurobi, etc) while focusing on expressing the logic of computation instead of its control flow description. Among the existing techniques, based on the paradigm of declarative programming and solving this type of problems, selected techniques of Constraint Programming (CP) exhibit a certain advantage that consists in accounting for the non-linear nature

of the problem. The CP techniques implemented here enable extending and adapting the reference model in question to other spheres of decision support that require the use of managerial decision-making support tools, for instance, designing the competency framework of academic staff, recruiting panels of experts for reviewing project applications, proposing variants of the composition of medical teams, etc. Therefore, given the range of applications and the capabilities of the presented method, it is a solution that shows good potential for implementation in project organisations susceptible to a range of potential disruptions, including unpredictable (at the planning stage) new orders, lack of resources to carry out the tasks, changing deadlines, etc. Further studies are planned to develop appropriate extensions of the current reference model and including solving larger-sized problems than currently considered cases, e.g., problem decomposition (dynamic programming) and/or involving techniques providing approximate solutions (Tabu search, population algorithms). Seeking to find time-effective solutions, the parallel research thread employs different variants of the hybrid approach [64] to implement the already developed model.

**Author Contributions:** Z.B. gave the theoretical and substantive background for the developed method, G.B. conceived and designed the experiments, J.W. prepared and provided mathematical description of the method, E.S. made an experimental verification of the method and A.G. provided technical guidance and gave critical review for this paper. All authors have read and agreed to the published version of the manuscript.

**Funding:** This research was funded by the National Science Centre, Poland, grant number 2019/33/N/HS4/00379.

**Conflicts of Interest:** The authors declare no conflict of interest.

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
