# Peer review of "Interactive Planning of Competency-Driven University Teaching Staff Allocation"

_applsci, doi:10.3390/app10144894_

Round 1

Reviewer 1 Report

The paper provides a new approach to the problem of qualified human resource allocation, which has many practical applications. The problem statement is interesting and novel. The structure of the manuscript is logical and clear. The paper has several very useful illustrations that help to understand the topic and the problem better.
At the same time there are some minor weaknesses that can be avoided:
1) The literature review is thorough, but it can be extended with some recent papers on the problem of resource allocation;
2) Please, include the discussion of limitations of the proposed method and compare it with alternative methods.
3) In the text, reference numbers should be placed in square brackets [ ]. Please, see the instructions for authors.
4) Line 216: a capital letter is needed

Author Response

We would like to thank You for very helpful, kind and careful comments and suggestions. We have incorporated all comments in the revised manuscript:

Remark 1: The literature review is thorough, but it can be extended with some recent papers on the problem of resource allocation;

Response: Thank you very much for the remark. We have added 6 additional recent references focusing on alternative approaches to resources allocation problem – see lines 150, 157 and 808, and in the bibliography – marked by green colour.

The new references are following:

  1. Burkova, I.; Titarenko, B.; Hasnaoui, A.; Titarenko, R. Resource allocation problem in project management. E3S Web of Conferences, 2019, 97, 01003, doi: 10.1051/e3sconf/20199701003.
  2. Nesterov, Y.; Shikhman, V. Dual subgradient method with averaging for optimal resource allocation, Eur. J. Oper. Res., 2018, 3(1), 907-916.
  3. Ivanova, A.; Pasechnyuk, D.; Dvurechensky, P.; Gasnikov, A.; Vorontsova, E. Numerical methods for the resource allocation problem in networks. Comput. Math. Math. Phys., 2020, https://arxiv.org/abs/1909.13321.
  4. Kłosowski, G.; Gola, A.; Świć, A. Application of Fuzzy Logic Controller for Machine Load Balancing in Discrete Manufacturing System. Lecture Notes in Computer Science, 2015, 9375, 256-263.
  5. Janardhanan, M.N.; Li, Z.; Bocewicz, G.; Banaszak, Z.; Nielsen, P. Metaheuristic algorithms for balancing robotic assembly lines with sequence-dependent robot setup times. Applied Mathematical Modelling, 2019, 65, 256-270.
  6. Tadic, I.; Marasovic, B. Analyze of human resource allocation in higher education applying integer linear programming. In: Beros M.B., Recker N., Kozina M. (Eds.) Economic and Social Development, 2018, 266-276.
  7. Sitek, P.; Wikarek, J. A multi-level approach to ubiquitous modeling and solving constraints in combinatorial optimization problems in production and distribution. Applied Intelligence, 2018, 48, 1344–1367.

Remark 2: Please, include the discussion of limitations of the proposed method and compare it with alternative methods.

Response: Thank you very much. Taking into account your suggestions and comments, we have supplemented the text of our contribution with the following fragments (pages 23-24):

“Although our method concerned employee absenteeism, it could be, in fact, applied to organisations experiencing other types of singular disruption, be it handling of additional, previously unplanned orders or changes in the duration of activities. In this context, the future work will explore the model capabilities to handle other disruptions that are commonly encountered in everyday practice. An important limitation of the adopted reference model is its discreet nature. In reality, the data are often of an uncertain, stochastic or fuzzy character (e.g. an approximately 5-day duration of an activity), which implies the need to take into account an appropriate, stochastic or fuzzy representation.”…

“Compared to existing methods [31, 44, 45, 46, 48, 49, 50, 51, 52, 53], the developed reference model copes with two types of inherent issues, that is the competency framework robustness analysis and the problem of competency framework synthesis robust to selected types of disruptions. The adopted paradigm of declarative modelling, unlike the currently available mathematical programming methods, allows direct implementation of the developed model in commercially available programming platforms (e.g. LINGO, IBM ILOG CPLEX, Gurobi, etc) while focusing on expressing the logic of computation instead of its control flow description. Among the existing techniques, based on the paradigm of declarative programming and solving this type of problems, selected techniques of constraint programming (CP) exhibit a certain advantage that consists in accounting for the non-linear nature of the problem.”

“Seeking to find time-effective solutions, the parallel research thread employs different variants of the hybrid approach [59] to implement the already developed model.”

Remark 3: In the text, reference numbers should be placed in square brackets [ ]. Please, see the instructions for authors.

Response: Thank you very much for the remark. We have made corrections in all references.

Remark 4:  Line 216: a capital letter is needed

Response: Thank you very much for pointing out the above-mentioned mistake. It has been corrected and marked by green colour (in line 209).

Reviewer 2 Report

The text is very well presented and argued in its methodological development as well as the objectives.

An explanation of how this situation or implementing this tool will benefit the quality of teaching is missing. There is no relationship to quality that should be important to include. The problem seems to be one of administrative planning and organization (e.g. in the case of teacher recruitment and its legal stability), for which no further details are given as to why the situation to be corrected is occurring.

It is not really clear in terms of the academic and teaching benefits that is the main objective of a school or educational organization.

Author Response

We would like to thank You for very helpful, kind and careful comments and suggestions. We have incorporated all the reviewers’ comments in the revised manuscript.

Reviewer 3 Report

Congratulations! The paper is well presented, the method is adequate. I am sure the paper will interest many, many readers :-)

Author Response

Thank you very much for your kind opinion.